# Characterization of the Human Eccrine Sweat Proteome—A Focus on the Biological Variability of Individual Sweat Protein Profiles

**DOI:** 10.3390/ijms221910871

**Published:** 2021-10-08

**Authors:** Bastien Burat, Audrey Reynaerts, Dominique Baiwir, Maximilien Fléron, Gauthier Eppe, Teresinha Leal, Gabriel Mazzucchelli

**Affiliations:** 1Mass Spectrometry Laboratory, MolSys Research Unit, Liège Université, B-4000 Liège, Belgium; g.eppe@uliege.be; 2Louvain Center for Toxicology and Applied Pharmacology (LTAP), Institut de Recherche Expérimentale et Clinique (IREC), Université Catholique de Louvain, B-1200 Brussels, Belgium; audrey.reynaerts@uclouvain.be (A.R.); teresinha.leal@uclouvain.be (T.L.); 3GIGA Proteomics Facility, Liège Université, B-4000 Liège, Belgium; d.baiwir@uliege.be (D.B.); m.fleron@uliege.be (M.F.)

**Keywords:** human eccrine sweat, shotgun proteomics, inter-individual variability

## Abstract

The potential of eccrine sweat as a bio-fluid of interest for diagnosis and personalized therapy has not yet been fully evaluated, due to the lack of in-depth sweat characterization studies. Thanks to recent developments in omics, together with the availability of accredited sweat collection methods, the analysis of human sweat may now be envisioned as a standardized, non-invasive test for individualized monitoring and personalized medicine. Here, we characterized individual sweat samples, collected from 28 healthy adult volunteers under the most standardized sampling methodology, by applying optimized shotgun proteomics. The thorough characterization of the sweat proteome allowed the identification of 983 unique proteins from which 344 were identified across all samples. Annotation-wise, the study of the sweat proteome unveiled the over-representation of newly addressed actin dynamics, oxidative stress and proteasome-related functions, in addition to well-described proteolysis and anti-microbial immunity. The sweat proteome composition correlated with the inter-individual variability of sweat secretion parameters. In addition, both gender-exclusive proteins and gender-specific protein abundances were highlighted, despite the high similarity between human female and male sweat proteomes. In conclusion, standardized sample collection coupled with optimized shotgun proteomics significantly improved the depth of sweat proteome coverage, far beyond previous similar studies. The identified proteins were involved in many diverse biological processes and molecular functions, indicating the potential of this bio-fluid as a valuable biological matrix for further studies. Addressing sweat variability, our results prove the proteomic profiling of sweat to be a promising bio-fluid analysis for individualized, non-invasive monitoring and personalized medicine.

## 1. Introduction

Sweat is secreted by millions of eccrine glands ubiquitously distributed over the human skin surface and involved in body thermoregulation by evaporation and skin homeostasis (hydration and immunity) [1,2]. Eccrine glands secrete a hypotonic solution of water-soluble electrolytes, e.g., sodium, chloride, potassium, urea or lactate and non-electrolytes, i.e., proteins, anti-microbial peptides and metabolites [3,4,5,6]. In clinical routine, investigations of sweat composition are limited to the determination of drug intake and the diagnosis of cystic fibrosis based on the measurement of sweat chloride concentration (Gibson and Cooke sweat test) [7,8].

Eccrine sweat is a promising bio-fluid of interest in the field of personalized medicine thanks to its less invasive collection relative to its complex composition, representing a valuable source of information [9] when compared to other routine bio-fluids such as bronchoalveolar, synovial or cerebrospinal fluids. In addition, the recent developments of high-resolution and high-sensitivity analytical techniques now allow working with small sweat volumes (10–20 µL) collected from standardized methods. Up to now, few proteomic studies have been conducted on eccrine sweat [10,11,12,13]. Despite the lower protein concentration (0.1–1 mg/mL), reports highlighted sweat as a likely informative source of biomarkers for monitoring both physiological and pathophysiological conditions; for example, disease-specific profiles of sweat proteins were described in patients with schizophrenia or tuberculosis [14,15]. However, all previous works were carried out on pooled sweat samples, thereby missing the inter-individual spectrum of sweat proteome variability.

In order to use sweat as a bio-fluid of interest, one must be able to characterize reference physiological profiles and inter-individual variations under steady-state conditions with standardized reproducible methods to discriminate between physiological responses to stimuli or disease-induced alterations. Eccrine sweat composition was described as complex and highly variable under normal physiological conditions, depending on multiple levels of stimulation and regulation, mechanisms of secretion and tissue contributions [16,17]. This complexity reflects the diversity of sweat’s biological functions, but it is also a major source of intra- and inter-individual variability.

In the context of personalized medicine, in light of the potential of eccrine sweat as a relevant source of protein biomarkers for prognosis or diagnosis of disease conditions and clinical follow-up, the current work was designed to carry out an in-depth characterization of the eccrine sweat proteome of healthy subjects. To this end, we applied standardized reproducible sweat collection, sample preparation, LC–MS/MS (liquid chromatography-tandem mass spectrometry) analysis and MS (mass spectrometry) data processing to the proteomic profiling of eccrine sweat. Nine hundred and eighty-three proteins were identified and quantified by a label-free approach across 28 individual samples, a significant improvement when compared to previous similar studies. Proteome annotation tackled newly described biological functions of sweat proteins and elaborated on the composition of sweat in terms of tissue contributions. In addition, the present study concluded on the effect of sweat rate on the inter-individual variability of sweat protein profiles and on the limited but significant influence of gender.

## 2. Materials and Methods

### 2.1. Sweat Collection

Sweat samples were collected from 30 healthy volunteers (HV, 15 female, 15 male) under the most standardized and spectroscopically pure conditions following the current recommendations for the Gibson and Cooke sweat test [18]. In brief, the volar region of both forearms was chosen based on its high density of eccrine sweat glands together with its very low density of apocrine/apoeccrine sweat glands and its easy access. Sweat samples were collected from each forearm successively, from fasting and well-hydrated individuals. Before sweat sampling, the tested skin region was washed with 70% ethanol, rinsed with ultrapure water and dried using ashless filter paper. Eccrine sweat secretion was stimulated by pilocarpine iontophoresis using pilocarpine gel-padded (Pilogel^®^ discs, ELITechGroup, Brussels, Belgium) electrodes: a 5 mA current (Webster Sweat Inducer, Model 3700, ELITechGroup, Brussels, Belgium) was applied for 5 min. After stimulation, the electrodes were removed and a Macroduct Sweat Collector (ELITechGroup, Brussels, Belgium) (blue dye removed with 70% ethanol and ashless filter paper) was attached in the place of the cathode to collect pure, non-diluted, pressure-driven eccrine sweat for 30 min. At the end of the collection time, the collector tubing was uncoiled, cut off and connected to a needle and syringe to transfer sweat to a 0.6 mL micro-tube.

### 2.2. Sample Preparation for Shotgun Proteomics

Pure, undiluted sweat samples were processed in three series of ten individual samples. Three sample preparation rounds (1 per series) were performed to avoid any technical bias that might come from a single sample preparation experiment

Sweat protein concentration was estimated using the Pierce Micro BCA™ Protein Assay kit (#23235, ThermoFisher Scientific, Waltham, MA, USA) according to the manufacturer’s instructions. Ten micrograms of proteins were precipitated by incubation in 90% acetonitrile for 30 min at 4 °C, followed by centrifugation for 10 min, at 4 °C, at 10,000× *g*. The protein pellet was re-suspended in 50 mM ammonium bicarbonate and incubated in: (i) 10 mM DTT (dithiothreitol) for 40 min at 56 °C, under stirring at 600 rpm (Thermomixer comfort, Eppendorf, Hamburg, Germany), to reduce disulfide bonds, (ii) 20 mM iodoacetamide protected from light for 30 min, at room temperature, to alkylate/block cysteine residues, (iii) 11 mM DTT protected from light for 10 min, at room temperature, to quench the residual iodoacetamide, (iv) mass-spectrometry-grade trypsin (Pierce™ Trypsin Protease, MS Grade, ThermoFisher Scientific, Waltham, MA, USA) at a 1:50 enzyme:protein ratio (protein concentration = 0.25 µg.µL^−1^), for 18 h at 37 °C, under stirring at 600 rpm (Thermomixer comfort, Eppendorf, Hamburg, Germany) and (v) MS-grade trypsin, at a 1:100 enzyme:protein ratio and acetonitrile to a final concentration of 80% (*v*/*v*), for 3 h at 37 °C, under stirring at 600 rpm (Thermomixer comfort, Eppendorf, Hamburg, Germany). Digestion was stopped by adding TFA (trifluoroacetic acid) to a final concentration of 0.5% (*v*/*v*). Samples were dried in a vacuum concentrator and re-suspended at 3.75 µg/20 µL in 0.1% TFA. At this step, aliquots from each sample were collected and mixed in three 10-sample pools, considering three series of 10 individual samples and an average pooled sample per series. Individual samples and pooled samples were desalted with C18 Zip Tips (Pierce™ C18 Tips, ThermoFisher Scientific, Waltham, MA, USA) according to the manufacturer’s recommendations, dried and re-suspended at 3 µg/9 µL (injection volume) in 0.1% TFA spiked with an equivalent of MassPREP Digestion Standard Mixture 1 (MPDS Mix 1, #186002865, Waters, Milford, MA, USA) corresponding to 50 fmol of ADH (alcohol dehydrogenase 1 from *Saccharomyces cerevisiae*) content per injection volume.

### 2.3. Liquid Chromatography and Mass Spectrometry Data Acquisition

The individual samples and pooled samples were sorted into three series of ten samples plus one pooled mix per series and analyzed using an ACQUITY UPLC M-Class liquid chromatography system (Waters, Milford, MA, USA) coupled with a Q-Exactive Plus Hybrid Quadrupole-Orbitrap mass spectrometer (ThermoFisher Scientific, Waltham, MA, USA). Three acquisition rounds (1 per series) were performed to avoid any technical bias that might come from a single LC–MS acquisition series.

The chromatographic separation consisted of a 3 min-long trapping step performed on a reversed-phase (RP) ACQUITY UPLC M-Class Trap Column (nanoEase MZ Symmetry C18 Trap Column, 100 Å, 5 μm, 180 μm × 20 mm, Waters, Milford, MA, USA) followed by a 177 min elution step on an ACQUITY UPLC M-Class Analytical Column (nanoEase MZ HSS T3 C18 Analytical Column, 100 Å, 1.8 μm, 75 μm × 250 mm, Waters, Milford, MA, USA) using a gradient of mixed water and acetonitrile, both supplemented with 0.1% formic acid, as eluents.

The mass acquisition was operated in data-dependent positive ion mode. Source parameters were set at: (i) 2.3 kV for spray voltage, (ii) 270 °C for capillary temperature and (iii) S-lens RF level = 50.0.

For individual samples, MS spectra were obtained for scans between *m*/*z* 400 and *m*/*z* 1600 with a mass resolution of 70,000 at *m*/*z* 200, an Automated Gain Control (AGC) of 3 × 10^6^, a maximum Injection Time (IT) of 200 ms and an internal lock mass calibration at *m*/*z* 445.12003. MS/MS spectra were obtained for the top 10 most intense ions of each MS scan (TopN = 10) with a mass resolution of 17.500 at *m*/*z* 200, an isolation window of 1.6 *m*/*z* with an isolation offset of 0.5 *m*/*z*, an AGC of 1 × 10^5^, a maximum IT of 200 ms and a (N)CE at 28. The exclusion of single-charged ions and a 10 s dynamic exclusion were enabled.

For each pooled sample, the MS acquisition consisted of a two-step strategy of three injections each. During both steps, MS spectra were obtained for scans between *m*/*z* 400 and *m*/*z* 528.3, *m*/*z* 524.3 and *m*/*z* 662.8 or *m*/*z* 658.8 and *m*/*z* 1600, in three independent analyses, respectively, with a mass resolution of 70,000 at *m*/*z* 200, an AGC of 3 × 10^6^, a maximum IT of 200 ms and internal lock mass calibrations at *m*/*z* 445.12003, *m*/*z* 536.16537 and *m*/*z* 684.20295, respectively. During the first acquisition step, MS/MS spectra were obtained for the top 25 most intense ions of each MS scan (TopN = 25) with a mass resolution of 17.500 at *m*/*z* 200, an isolation window of 1.6 *m*/*z* with an isolation offset of 0.5 *m*/*z*, an AGC of 1 × 10^5^, a maximum IT of 250 ms and a (N)CE at 28. For the second acquisition step, an exclusion list for all signals related to peptides identified at the first step with more than 4 PSM (peptide-spectrum matches) was uploaded to the methods. During the second acquisition step, MS/MS spectra were obtained for the top 10 most intense ions of each MS scan (TopN = 10) with a mass resolution of 17.500 at *m*/*z* 200, an isolation window of 1.6 *m*/*z* with an isolation offset of 0.5 *m*/*z*, an AGC of 1 × 10^5^, a maximum IT of 600 ms and a (N)CE at 28. The exclusion of single-charged ions and a 15 s dynamic exclusion were enabled for both steps. 

### 2.4. Bioinformatic Analysis and Statistics

Raw MS data were submitted to protein identification and label-free quantification by the MaxQuant software (version 1.6.6.0) using default settings when not specified otherwise. Identification consisted of a search against a custom-made reviewed Uniprot *Homo sapiens* database (20443 *Homo sapiens* entries + 4 MPDS Mix 1 entries, release date 8 August 2019) with carbamidomethyl (C) set as a fixed modification, oxidation (M) and deamidation (NQ) set as variable modifications and a minimum of two peptides (including one unique peptide) required. LFQ (label-free quantification) was enabled with a minimum LFQ ratio count of 1, no Fast LFQ and no requirement of MS/MS for LFQ comparison. The ‘match between runs’ option was enabled and tuned to allow matches from the library (pooled aliquots considered as parameter group 1, ‘match from’) and between individual samples (parameter group 0, ‘match from and to’). A match time window of 2.5 min was used.

MaxQuant output data (proteinGroups.txt) were submitted to statistical analysis using the Perseus software (version 1.6.10.43). ‘Only identified by site’, ‘REVERSED’ and contaminant data were filtered out. LFQ intensities were log2-transformed and proteins with less than 50% of valid values were filtered out. Principal component analysis (PCA) was performed on Z-score-normalized LFQ intensities. The male versus female comparison was addressed by a two-sample Student’s t-test with a permutation-based FDR calculation. Proteins with a fold change above 2 and an FDR below 0.05 were considered significantly differently expressed between female and male groups. Heat-map representation and hierarchical clustering were generated using the average Euclidian distance calculation.

PERMANOVA (PERmutational Multivariate ANalysis Of VAriance) tests were performed with the PAST software (version 4.04) using the hierarchical clustering distance matrix with the number of permutations set to 999.

The classification of identified proteins and over-representation/enrichment were conducted using the online search engine powered by the PANTHER Classification system. The PANTHER Overrepresentation test (release date 28 July 2020) parsed the PANTHER database (version 10.5281/zenodo.4081749, release date 9 September 2020) using the Homo sapiens reference list, the PANTHER-Gene Ontology-Slim and PANTHER Protein Class annotation datasets. Only *p* < 0.05 items were retained and considered significantly over-represented.

Visualization of proteome overlaps was achieved by submitting SwissProt accession numbers to the online Venn diagram generator Venny (version 2.1.0, https://bioinfogp.cnb.csic.es/tools/venny, access date 10 November 2020).

Visualization of interaction networks was achieved by submitting SwissProt accession numbers to STRING (version 11.0, https://string-db.org, access date 27 April 2021).

Due to the limited and variable sweat volume collected, together with the relatively low and variable sweat protein concentration plus the need to store sweat samples for further analyses, no technical replicate was performed. To account for technical variability, sweat samples were processed and analyzed in three series of ten individual samples.

### 2.5. Subject Inclusion and Study Approval

All 30 sweat samples were collected at steady-state conditions from volunteers with no known acute or chronic illness, no known drug use at the time of collection, no cosmetic uses or skin damage at the site of collection and no clinical sign of dehydration. Subjects were asked to fast and keep well hydrated over a minimum of 8 h before collection. Female volunteers were neither pregnant nor lactating. The study was approved by the ethics committee of Cliniques Universitaires Saint-Luc—Université Catholique de Louvain faculty hospital (ClinicalTrials.gov identifier: NCT03993600). All subjects signed an informed consent and were not enrolled in another clinical trial. A material transfer agreement was signed with Université de Liège for sample analysis.

After sweat collection, sweat chloride (coulometry, ChloroChek chloridometer, ELITechGroup, Brussels, Belgium), sodium and potassium concentrations (flame photometry, Flame Photometer Model 420, Sherwood Scientific, Cambridge, UK) were measured. All samples presented physiologically normal values.

## 3. Results

### 3.1. The Proteome of Human Eccrine Sweat

Individual eccrine sweat samples were analyzed separately to gain insight into their individual proteome. To allow optimal protein identification, missing identifications (IDs) from individual analysis were completed with IDs transferred from the analysis of both other samples in the cohort and pooled aliquots thanks to the match between runs option of the MaxQuant software (version 1.6.6.0) (Figure 1).

### 3.2. Protein Identification: Comparison to Previous Studies

Sweat samples from 30 healthy subjects—see Appendix A for a complete clinical data summary—were analyzed by nanoLC–MS/MS. Based on chromatogram discrepancy, poor correlation with the other sample data and a low number of identified proteins, two samples were discarded (Appendix A)—clinical data related to the 28 remaining samples are summed up in Table 1. Eventually, considering a minimum of two peptides—including one unique peptide—and a False Discovery Rate (FDR) below 0.01 for protein identification, a total of 986 proteins were identified, accounting for data filtering (Appendix A) and the standard protein mixture (MPDS Mix 1, Waters) for quality control. A total of 535 ± 22 proteins were peptide-spectrum matching hits, while 273 ± 9 proteins required matching between runs for identification for an average total of 808 ± 16 proteins identified for each sample (Figure 2a). A total of 347 proteins were consistently identified across all samples.

The optimization of sweat sample preparation and MS acquisition together with the strategy for thorough protein identification significantly improved the analysis depth, as shown by the comparison with earlier sweat proteomics reports (Figure 2b). In terms of proteome coverage, the overall number of identified proteins was significantly higher when compared to previous studies with similar workflows. In addition, the increase in protein identification was characterized by an extended overlap with previously reported proteomes. It is noteworthy that while those studies relied on the use of pooled sweat samples to account for the limited volume available, the current work focused on the analysis of individual samples, with similar or better performance in protein identification.

### 3.3. Protein Classification and Over-Representation

Identified sweat proteins were classified and tested for over-representation by the PANTHER Classification system and Gene Ontology Enrichment analysis, mapping protein IDs against PANTHER GO Slim annotation datasets (Appendix A). The sweat proteome was significantly enriched in: (i) proteins related to proteolytic activity, proteases and peptidases [11] as well as their respective inhibitors, and (ii) protein effectors and regulators of the innate and adaptive immune systems [10,13]. Moreover, (i) cytoskeletal proteins, i.e., protein components (actin and ABP) of the actin cytoskeleton and regulators of actin organization and dynamics, (ii) proteins of reactive oxygen species metabolism and oxidative stress, (iii) markers of UPR and ER stress, (iv) components and regulators of the proteasome or (v) proteins of all major metabolic pathways, were among the over-represented proteins mapped to annotation clusters. A rapid annotation of the interaction network of sweat core proteins highlighted interaction clusters related to the biological functions mentioned above (Figure 3).

### 3.4. Relative Contributions of Plasma and the Eccrine Gland to Sweat Protein Composition

When comparing the 20 most abundant proteins of sweat (from the current study, 2.8% of total sweat proteins) to the plasma proteome [19] (Figure 4a), only 13 sweat proteins were retrieved in plasma and 7 were exclusive to sweat. Conversely, when comparing the 20 most abundant proteins of plasma to the sweat proteome, only 11 plasma proteins were retrieved in sweat, 9 being exclusive to plasma. Proteins exclusive to sweat, together with the absence of proteins shared between the top 20 proteins of plasma and sweat, highlighted the specificity of the sweat proteome. This observation excluded the filtration of plasma as the sole contributor to sweat protein composition. On a side note, the covered dynamic range of sweat was quite similar to that of plasma, spanning five orders of magnitude (Figure 4b).

The overlap of proteins between the current work and earlier plasma and eccrine gland proteomic reports was estimated so as to evaluate the different contributions to the sweat proteome at a larger scale (Figure 4c). Comparison to plasma and eccrine gland proteomes showed that 19.7% of sweat proteins were specific to sweat while 6.5% were shared between sweat and plasma, 44.8% were shared between sweat and the eccrine gland and 29% were shared between the three proteomes of interest.

Mapping proteins of different tissue origins to PANTHER-GO annotation clusters, the over-representation test determined that: (i) proteins exclusive to the current sweat proteome dataset were predominantly mapped to annotation clusters related to proteolytic activity and immune systems, (ii) proteins of plasma origin were mapped to annotation clusters related to proteolytic activity, immune systems, oxidative stress, proteasome and metabolic pathways, (iii) proteins originating from the eccrine gland were mapped to annotation clusters for proteolytic activity, actin organization and dynamics, proteasome and metabolic pathways and (iv) proteins of mixed origins were mapped to annotation clusters for proteolytic activity, actin dynamics, oxidative stress, UPR and ER stress, proteasome and metabolic pathways (Appendix A).

In conclusion, sweat is not a transudate of plasma, due in part to the active input of the eccrine gland in sweat protein composition.

### 3.5. Inter-Individual Variability of the Sweat Proteome

A total of 986 identified proteins were suitable for protein label-free quantification. For further statistical analysis of protein abundances across samples, only those identified and quantified in at least 50% of the samples (14 out of 28) were used, for a total of 873 proteins.

First, the combined biological and technical variability was estimated by calculating the coefficients of variation (CV) of protein abundances measured across the 28 remaining samples. The distribution of combined biological and technical CV was density-plotted against the median log10-transformed protein abundance, as shown in Figure 5a. The variability across sweat samples was higher than the variability across plasma samples [19] and was on par with the variability across urine samples [20]. A total of 639 proteins had a CV below 100%, of which 105 proteins had a CV below 50%.

Computation of Pearson’s correlation coefficients (PCC) and average Euclidian distance hierarchical clustering were used to classify samples according to profiles of pairwise correlation with other samples in the cohort. Samples were sorted into nine distinctive clusters of significant sample correlation, i.e., sweat proteome profiles according to the joint observation of the clustering tree structure and the “heat” of PCC (Figure 5b). Available clinical data (including age, gender, BMI plus collected sweat volumes, ion concentrations, ion amounts, protein concentration and protein mass) were tested for their correlation with the biological inter-individual variability of the sweat proteome.

First, numeric clinical data were color coded based on their relative position in the quartile-delimitated distribution (Appendix A) to estimate if hierarchical clustering had grouped samples with matching clinical parameters. Anthropometric parameters such as age, gender and BMI did not appear as likely contributors to the inter-individual variability of the sweat proteome, as clusters did not group samples with matching age, gender or BMI. However, specific sweat-related parameters such as collected sweat volume (water loss), ion molarities and ion amounts (ion loss) as well as protein concentration and protein mass (protein loss) likely reflected the variability of sweat proteome profiles, since hierarchical clustering grouped samples with matching levels of water, ion or protein loss. In other words, variations in such sweat-related parameters might translate to variations in sweat proteome profiles.

Then, anthropometric and sweat-related parameters were tested for the significance of their correlation with the clusters of sample correlation by PERMANOVA (Figure 5b, lower panel), which compared clinical data distribution to hierarchical clustering in an objective way. As expected, no significant correlations between anthropometric parameters (age, sex and BMI) and sweat proteome profiles were observed. Neither were significant correlations between sweat-related parameters (right and left arm collected volumes, Na^+^, Cl^−^ and K^+^ concentrations, Cl^−^ and K^+^ amounts and protein mass) and sweat proteome profiles. Only secreted Na^+^ distribution stood out as significantly correlated with sweat proteome profiles and was emphasized as a core byproduct of the inter-individual variability of the sweat proteome.

To sum up, only the local sweat-related parameter, secreted Na^+^, but not anthropometric parameters, was a significant marker of the biological inter-individual variability of sweat protein composition, at steady-state conditions upon pilocarpine iontophoresis.

### 3.6. Inter-Gender Variability of the Sweat Proteome

Assessing the inter-gender variability of sweat proteins, the differential composition of male and female sweat proteomes was first estimated: 99.4% of sweat proteins were shared between male and female volunteers. Of 986 identified proteins, only 4 proteins—catenin α-1 (2/13 female HV), glutathione S-transferase θ-2B (4/13 female HV), immunoglobulin κ variable 2–28 (2/13 female HV) and leucine-rich repeat-containing protein 15 (1/13 female HV)—were exclusive to female HV, and two proteins—prostate-specific antigen (4/15 male HV), V-set and transmembrane domain-containing protein 2A (8/15 male HV)—were exclusive to male HV (Figure 6a). Furthermore, while principal component analysis between female and male samples did not indicate any gender-related dispersion (Figure 6b), significant changes in protein abundance between male and female sweat were observed for 3 out of 986 proteins (Figure 6c). Kallikrein-11 (log_2_ LFQ (Male − Female) = −2.43, *p* = 5.64 × 10^−7^, *p* < 0.001^***^, 28/28 HV) was significantly more abundant in female sweat while prolactin-inducible peptide (log_2_ LFQ (M − F) = 1.97, *p* = 1.55 × 10^−4^, *p* < 0.001 ^***^, 28/28 HV) and signal peptide, CUB, EGF-like domain-containing protein 2 (log_2_ LFQ (M − F) = 2.48, *p* = 9.77 × 10^−7^, *p* < 0.001 ^***^, 26/28 HV) were significantly more abundant in male sweat (Figure 6d).

## 4. Discussion

The current study was designed to run a thorough and in-depth characterization of the eccrine sweat proteome of healthy subjects, at an individual level. This was achieved with 983 sweat proteins (including 344 core proteins) identified and quantified by inter-individual relative abundance, while previous results from published studies were obtained through the analysis of pooled individual samples. A standardized and optimized workflow, from the sample collection and sample preparation to the LC–MS analysis and the bio-informatics for data processing, was developed. Altogether, the present methods and results described above could be considered a reference for further personalized monitoring studies.

Over the past decade, eight studies have focused on the proteomic profiling of sweat. To do so, four different sweat collection strategies were tested: (i) pharmacological cholinergic stimulation by pilocarpine iontophoresis, at a volar forearm site, and collection using the Macroduct system [14,15], (ii) heat stimulation and collection [10], (iii) stimulation by physical exercise and collection using absorbing tissue pads at multiple body sites [11] and (iv) stimulation by physical exercise and collection with the Macroduct system [12,13].

All studies opting for Macroduct collection resulted in a low depth of proteome coverage regardless of the stimulation method. Such a low yield can be explained by low collected volumes, given that, in the present work, sweat volumes two to three times larger than those reported in previous studies (collected over the same 30 min interval) resulted in a five to twenty times larger number of identified proteins (in the same conditions of stringency). Interestingly, before the present study, the largest reported number of identified proteins was obtained by doubling the duration of Macroduct collection [13]. The importance of sweat was confirmed when Yu et al. [11] chose a different collection method, using absorbing tissue pads at different body sites to collect larger volumes. However, it is noteworthy that the increase in the collected volume of sweat was hindered by the greater probability of skin protein contamination.

By following gold standard consensus guidelines for sweat testing [18], the current work ensured highly standardized and reproducible sweat stimulation, sampling and handling, under spectroscopically pure conditions, preventing issues related to contamination and sweat evaporation. On top of these recommendations, the collection from fasting, well-hydrated, healthy subjects set a steady state for variations in biological origin, and limited protein identification and quantitation biases. For the first time, standardized reproducible sweat samples from healthy subjects were individually analyzed by optimized, reproducible shotgun proteomics with high-resolution LC–MS/MS acquisition. Therefore, the current study achieved the most thorough characterization of the eccrine sweat proteome to date, setting references for protein composition and individual proteome profiles of eccrine sweat.

The earliest reports of sweat proteomics already described a significant over-representation of proteolysis markers, together with markers of the immune system [10,11]. In line with the role of sweat in skin homeostasis and defenses, these results highlight an extended set of proteases, peptidases and their inhibitors regulating the metabolism of anti-microbial peptides in the broader context of skin innate immunity [21,22,23,24]. Protein families newly described in the context of sweat, such as cytoskeleton, oxidative stress, UPR and ER stress, proteasome and metabolism proteins, indicated a far more complex bio-fluid in terms of the diversity of biological functions. This diversity can be explained by the presence of cell exosomes in sweat [25].

The comparison between sweat, eccrine glands and plasma proteomes helped precise tissue contributions in terms of protein numbers and biological functions. First, the current work confirmed sweat did not result from the passive transport from plasma through the eccrine gland cells. Both plasma and eccrine gland cells contribute to the sweat proteome, as shown by proteins of plasma, eccrine or plasma/eccrine origins. Moreover, the contribution of the eccrine gland proteome seemed the most dominant.

As discussed above, the depth of proteome coverage helped focus on non-described sweat protein functions. The comparison to plasma and eccrine gland proteomes showed that the diversity of biological functions of proteins from a given origin did not reflect the diversity of biological functions of sweat proteins as a whole. For instance, proteins related to actin dynamics preferentially came from eccrine gland cells, while immune system proteins preferentially originated from plasma.

A significant part of the sweat proteins was not of plasma or eccrine origin. Other sources of proteins could be exosomes of non-eccrine cells and immune system cells or protein contamination from corneocytes at the collection site, despite the fact that a far less contamination-prone collection method was chosen. It is noteworthy that shotgun proteomics studies are not exhaustive in terms of proteome comprehension, by nature, so, our knowledge of sweat protein origins is bound to evolve with more thorough studies to come.

Beyond the need for a more in-depth coverage of the sweat proteome, the current work aimed at a better understanding of the inter-individual variability of sweat proteome profiles. In the context of heat acclimation and aerobic-exercise-related conditions, anthropometric parameters such as age, sex and body mass index were identified as notable sources of the intra- and inter-individual variability of sweat secretion [16,17], e.g., the onset and rate of sweating, with little evidence of age-, sex- or BMI-modified sweat composition. Sweat rate was described as the main contributor to the variability of sweat composition, and all age, sex or BMI effects on sweat composition were byproducts, i.e., sweat rate effects beforehand. Therefore, for example, Na^+^ and Cl^-^ sweat concentrations directly correlated with sweat rate.

Sweat protein composition was correlated with sweat rate, as protein content increased with increasing sweat rate, in heat acclimation experiments [26]. At steady-state conditions, upon pharmacological sweat stimulation (pilocarpine iontophoresis), the present study confirmed observations from heat acclimation and aerobic exercise: at steady state, (i) the anthropometric parameters age, sex and BMI did not contribute to the inter-individual variability of the sweat proteome, and (ii) sweat-rate-related parameters and sweat composition were correlated with the inter-individual variability of the sweat proteome, Na^+^ secretion being the main variable of interest.

The importance of gender must be discussed further. Conflicting reports about the inter-gender variability of the sweat proteome have arisen in the past few years. Yu et al. [11] concluded that gender-specific biomarkers were found in sweat, while Harshman et al. [12] determined that male and female sweat were similar in terms of protein composition. Although gender did not influence the inter-individual variability of the sweat proteome as a whole, it appeared there were minor gender-related differences, highlighted by a few gender-specific differential protein abundances. These results reconcile previous reports [11,12], since limited gender-specific biomarkers seem to exist despite a near complete similarity.

In conclusion, standardized sample collection coupled with optimized shotgun proteomics significantly improved the depth of sweat proteome coverage, far beyond previous similar studies. Addressing the inter-individual variability of the sweat proteome, our results demonstrate the proteomic profiling of sweat to be a promising lead in the search for non-invasive, individualized monitoring of protein biomarkers in relation to biometric tracking, clinical follow-up and personalized medicine. In this regard, investigators should pay particular attention to inter-individual variations in sweating rate.

## Figures and Tables

**Figure 1 ijms-22-10871-f001:**
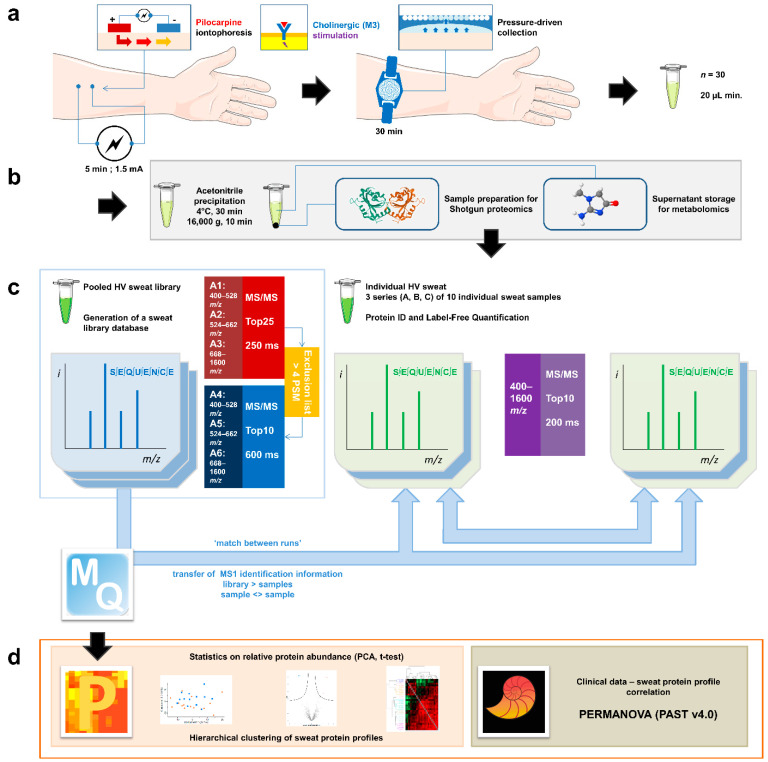
General experimental workflow. (**a**) Standardized sweat collection method. (**b**) Single sample preparation method for subsequent sweat proteomics and metabolomics studies. (**c**) Analytical and bioinformatic strategy using the “match between runs” option (MaxQuant). 1. Generation of a sweat reference proteome database from pooled samples. A two-round analysis of three limited *m*/*z* range acquisitions was performed. A precursor exclusion list was applied for the second round experiment. 2. Individual sweat sample analyses. (**d**) Statistical data processing tools.

**Figure 2 ijms-22-10871-f002:**
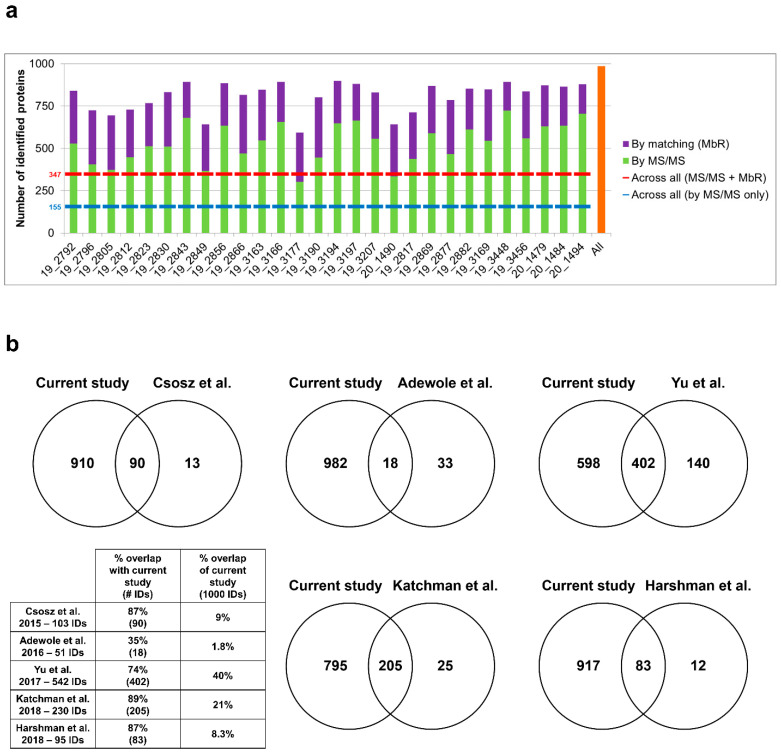
Performance of sweat shotgun proteomics. (**a**) Number of proteins identified (min. 2 peptides, 1 unique peptide) in each sample, by MS/MS (**in light green**) and by MbR (**in purple**), with total number of identified proteins (**in orange**). Standard proteins (*n* = 3, MPDS Mix 1) were kept as quality control. (**b**) Protein identification overlap with previous sweat proteomics studies (min. 2 unique peptides). Contaminant proteins were kept for proteome comparison. Here, the reference proteome included 975 proteins of interest plus 3 standard proteins (MPDS Mix 1) and 22 contaminants, identified with a minimum of 2 unique peptides.

**Figure 3 ijms-22-10871-f003:**
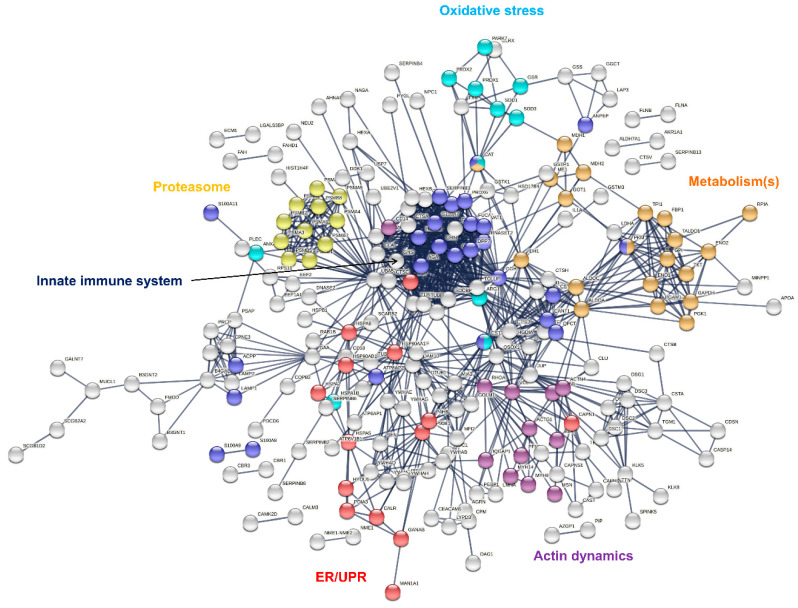
Interaction network mapping of sweat core proteins (347 proteins in 28/28 samples). The 347 query proteins resulted in 337 mapped proteins. Network settings included: *Homo sapiens* database, highest confidence minimum required interaction score, hidden disconnected nodes and confidence-based networking.

**Figure 4 ijms-22-10871-f004:**
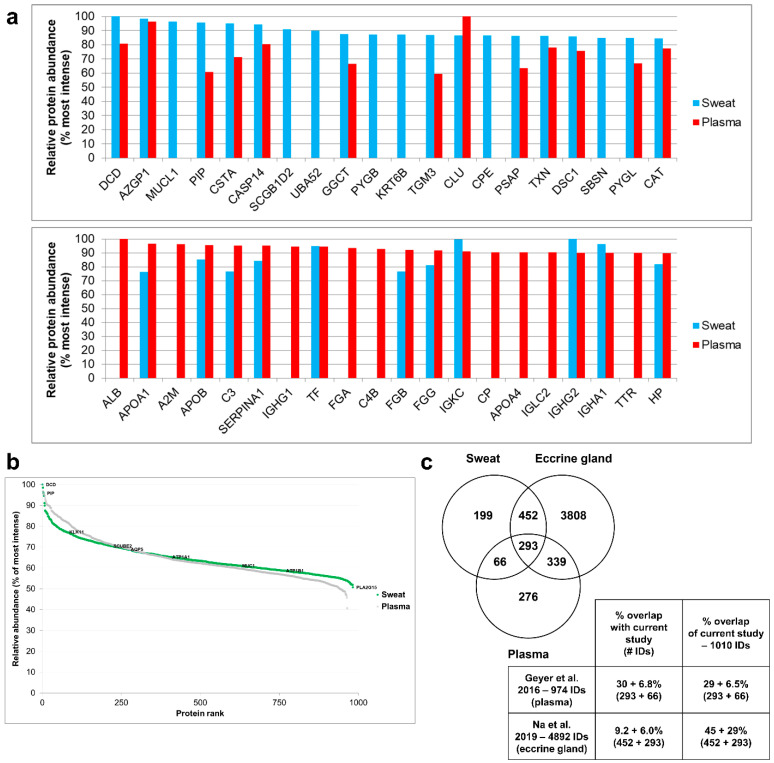
Tissue contributions to sweat composition. (**a**) Most abundant proteins (top 20) of sweat, as found in the plasma proteome (**upper panel**); most abundant proteins of plasma, as found in the sweat proteome (**lower panel**). Relative protein abundance = % Log10 LFQ intensity of top 10. (**b**) Dynamic range of sweat compared to plasma. Most and least intense proteins as well as eccrine gland markers in sweat were annotated. (**c**) Protein identification overlap with plasma and eccrine cell lysate (all identified proteins). Contaminant proteins were kept for proteome comparison. Here, the reference proteome included 983 proteins of interest plus 3 standard proteins (MPDS Mix 1) and 24 contaminants, identified with a minimum of 2 peptides, including a minimum of 1 unique peptide. Plasma proteomic data were retrieved from Geyer et al. 2016. Eccrine gland cell proteomic data were retrieved from Na et al. 2019.

**Figure 5 ijms-22-10871-f005:**
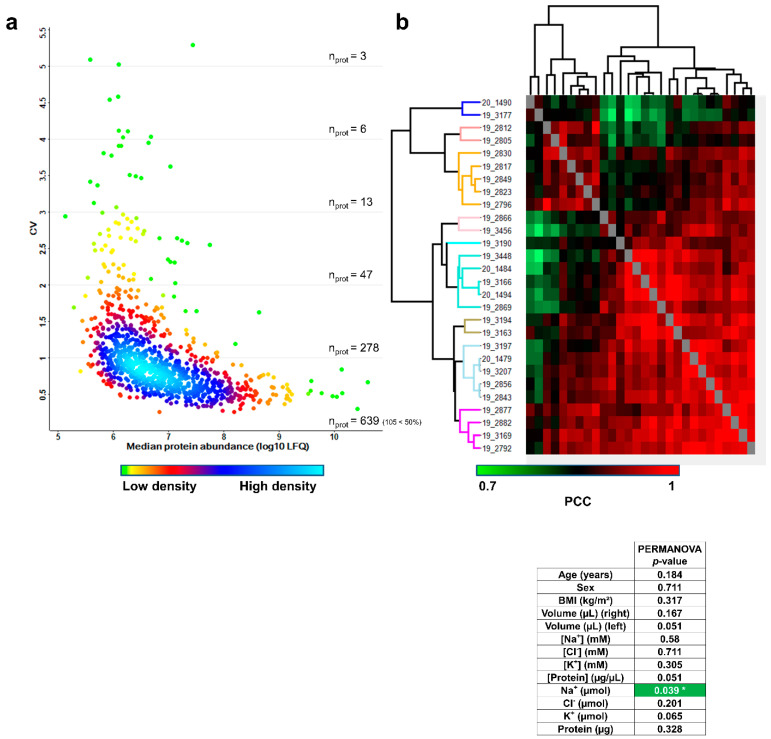
Inter-individual variability of the sweat proteome. (**a**) Coefficient of variation (CV) distribution (**b**). Heat-map representation of 28 sweat protein profiles. Hierarchical clustering of Pearson’s correlation coefficients generated using the average Euclidian distance matrix. PERMANOVA test for significance of correlation between clustering and clinical data distribution (number of permutations = 999, significant correlation highlighted in green, *p* < 0.05 *).

**Figure 6 ijms-22-10871-f006:**
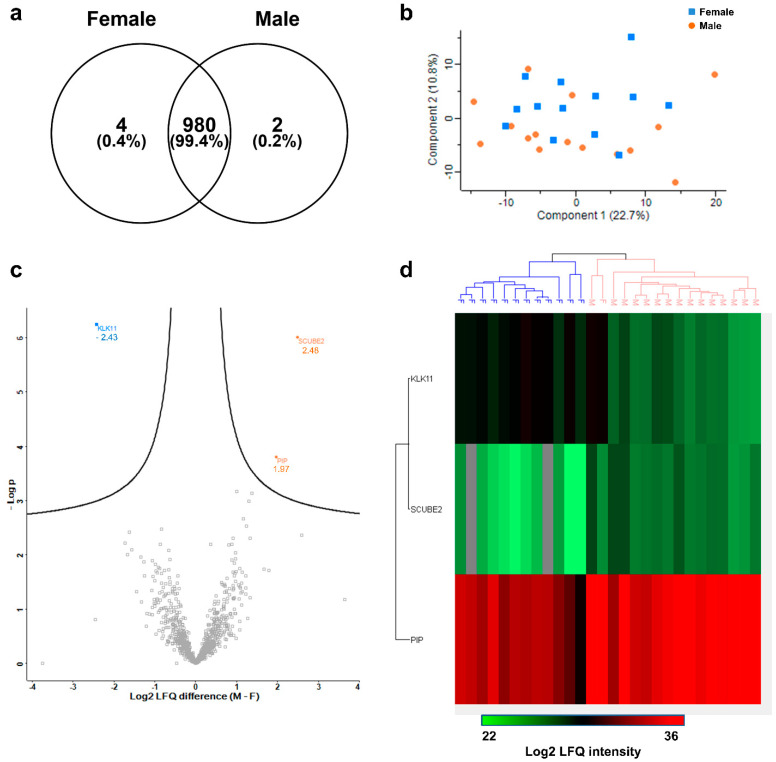
Inter-gender variability of the sweat proteome. (**a**) Male/female proteome overlap. Standard proteins (*n* = 3, MPDS Mix 1) were kept as quality control. Here, the sweat proteome included 983 proteins of interest plus 3 standard proteins (MPDS Mix 1), identified with a minimum of 2 peptides, including a minimum of 1 unique peptide. (**b**) PCA, male subjects (**plain orange dots**) and female subjects (**plain blue squares**). (**c**) Volcano plot. (**d**) Heat-map representation of gender-specific sweat protein abundances.

**Table 1 ijms-22-10871-t001:** Clinical data summary.

Sample IDs	Age (years)	Sex	BMI (kg/m²)	Collected Volume (µL) (Right Arm)	Collected Volume (µL) (Left Arm)	[Na^+^] (mM)	[Cl^−^] (mM)	[K^+^] (mM)	[Protein] (µg/µL)	Na^+^ (µmol)	Cl^−^ (µmol)	K^+^ (µmol)	Protein (µg)	
19_2792	36	M	22.7	62.1	87.8	56	26	8	0.488	3.48	1.61	0.50	42.85	HV (*n* = 28)
19_2796	29	M	31.6	48.1	42.4	23	4	10	0.557	1.11	0.19	0.48	23.62	
19_2805	28	M	30.0	98.9	97.5	59	24	7	0.340	5.84	2.37	0.69	33.15	
19_2812	40	M	25.5	89.6	96.5	39	18	8	0.432	3.49	1.61	0.72	41.69	
19_2866	74	M	21.3	56.9	64.2	37	12	12	0.551	2.11	0.68	0.68	35.37	
19_2823	31	F	19.8	71.8	89.8	32	10	10	0.262	2.30	0.72	0.72	23.53	
19_2830	30	F	23.1	94.0	77.9	32	11	8	0.462	3.01	1.03	0.75	35.99	
19_2843	24	F	18.2	63.5	64.1	65	36	10	0.472	4.13	2.29	0.64	30.26	
19_2849	27	F	20.5	97.7	84.6	22	10	8	0.432	2.15	0.98	0.78	36.55	
19_2856	29	F	18.7	79.7	70.4	28	6	8	0.422	2.23	0.48	0.64	29.71	
19_3163	32	F	19.4	44.8	30.1	30	12	9	0.785	1.34	0.54	0.40	23.63	
19_3166	26	F	18.9	57.2	64.0	44	22	10	0.494	2.52	1.26	0.57	31.62	
20_1490	39	F	21.3	40.1	43.5	26	12	8	0.371	1.04	0.48	0.32	16.14	
19_3177	28	M	23.1	35.9	27.0	51	30	8	0.475	1.83	1.08	0.29	12.83	
19_3190	29	M	24.5	98.6	80.7	55	28	6	0.406	5.42	2.76	0.59	32.76	
19_3194	41	M	24.1	34.4	42.7	36	8	9	0.736	1.24	0.28	0.31	31.43	
19_3197	36	M	22.5	27.5	82.7	91	44	8	0.504	2.50	1.21	0.22	41.68	
19_3207	28	M	23.8	84.9	93.2	45	22	7	0.364	3.82	1.87	0.59	33.92	
20_1494	29	F	20.9	61.3	64.4	54	30	12	0.441	3.31	1.84	0.74	28.40	
19_2869	28	F	17.7	57.1	45.9	67	10	24	0.646	3.83	0.57	1.37	29.65	
19_2877	33	F	22.3	102.7	83.7	35	12	7	0.350	3.59	1.23	0.72	29.30	
19_2882	24	F	20.6	57.2	53.6	29	10	6	0.377	1.66	0.57	0.34	20.21	
19_3169	57	F	19.2	35.6	29.9	58	24	8	0.661	2.06	0.85	0.28	19.76	
20_1479	25	M	24.5	56.4	72.6	69	38	7	0.336	3.89	2.14	0.39	24.39	
20_1484	24	M	23.6	99.0	93.4	49	20	8	0.622	4.85	1.98	0.79	58.09	
19_2817	26	M	20.8	47.4	40.4	73	44	9	0.314	3.46	2.09	0.43	12.69	
19_3448	30	M	20.6	91.2	74.0	50	22	10	0.448	4.56	2.01	0.91	33.15	
19_3456	31	M	22.6	101.5	99.5	69	40	5	0.198	7.00	4.06	0.51	19.70	

Mean	33	13 F 15 M	22.2	67.7	67.7	47	21	9	0.462	3.13	1.39	0.58	29.72	
Median	29		21.8	61.7	71.5	47	21	8	0.445	3.16	1.22	0.59	29.98	
SD	11		3.2	24.2	22.7	18	12	3	0.137	1.49	0.89	0.24	9.87	
SEM	2		0.6	4.6	4.3	3	2	1	0.026	0.28	0.17	0.05	1.87	

## Data Availability

The MS proteomic data have been deposited into the ProteomeXchange Consortium (http://proteomecentral.proteomexchange.org) [27] via the PRIDE partner repository [28] with the data set identifiers PXD025899 and 10.6019/PXD025899.

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
