# Peer review of "Characterization of the Human Eccrine Sweat Proteome—A Focus on the Biological Variability of Individual Sweat Protein Profiles"

_ijms, 2021, doi:10.3390/ijms221910871_

Round 1
Reviewer 1 Report
The paper by Burat et al represents the most comprehensive eccrine sweat proteome which is known in the field. Despite their descriptive nature, projects of this type are of the high value as they provide catalogue of protein species of a specific human fluid, which then may be referenced by further studies more relevant to the human disease. I highly appreciate this well-planned and well-described study which was done on a state-of-the-art technical level. Not many serious notes can be made to improve the paper, however there are some issues.
- Pathway enrichment analysis of a whole proteome is not very informative. In addition to what is done, please consider providing enrichments for the subsets of sweat-specific proteins, especially, for those which are identified at least at two individuals.
- Table 1 with clinical parameters can be cut down. Four right columns seem to be excessive.
- Page 11. A first paragraph on this page is difficult for understanding, please rephrase it.
In summary, I think this work is done very professionally and thoroughly. It does not contain any long narrative, but, however, represents publicly available, reliable data very useful for the community.
Reviewer 2 Report
The manuscript presents detailed studies about the characterization of the human eccrine sweat proteome. The study is designed, executed, and described in a sufficient manner. I have only minor comments:
1) add to supplement FASTA files with sequences (both peptides and proteins) as separate files (ideally, one would like to get 28 (or even 30) files where she/he can find: peptides, proteins associated with them - both IDs and sequences) - even though it may seem redundant to the deposition in PRIDE repository.
2) separate sheets from xlsx supplementary file could be divided into separate CSV or/and JSON files
For instance "Identified Proteins" sheet has multiple non-atom cells with content separated by ";" - you have multiple proteins/peptides assigned to the same row/cell. This is related to the nonuniqueness of peptides vs proteins, but, as you did not include peptides and protein sequences, it will be almost impossible or extremely cumbersome to go back and search for peptides that are shared by, for instance 5, mentioned proteins. Additionally, in the future, the protein sequences in UniProt or any other database will be updated (e.g. they will be obsolete, replaced by newer versions, etc.). Therefore, I recommend you to add to this sheet also columns with the peptide and protein sequence you are referring to.
In general, xlsx is an extremely poor choice of data format (I know that it is frequently preferred by publishers, or it can be even required to use xlsx), but it should not stop you from mirroring the supplementary data in some more proper formats (e.g. to avoid the problem with "one to many" relations, you could use JSON format).
3) The manuscript needs some proofreading by a native speaker and rewriting some parts would be also beneficial. Currently, you can find multiple problems with grammar, typos, and punctuation. Just to name a few:
- in abstract "medicine.Here"
- punctuation:
a) page 3 "incubated in : i)" should be "incubated in: i)"
b) page 9 "in : i) proteings"
c) page 9 "Moreover,:"
etc.
- Legend for Fig. 3 from "347 query proteins, ..." make no sense, those are not even sentences
- page 11, the first half of the first paragraph of "3.4. Relative ..." (fragment "Evaluating the abundance ... (Figure 4b)") is extremely hard to follow (should be rewritten)
- never use contractions (like "didn't") in the writing
- "contamination & sweat"
- abstract:
a) "about 1000 proteins" -> state exact number, there is no room for "about" in a science paper when you refer to the main result of your work, especially in the abstract (the same should be done for the rest of the text)
b) "appeared to be correlated" -> either it is or not, it can be also moderate, strong, etc. You could not find a worse description for correlation than "appeared to be", this is like you would doubt in any result in your work
4) There is quite a big problem with the number of proteins shown in the figures. For instance, Fig 1b Ven diagrams contain 1000 proteins, in 4c you have 1010, in 6a you have 986. The explanation for the last one I could find in the text, but from where you got 1000 or even 1010 proteins? You should explain the differences in protein counts in tables' legends or correct the numbers (additionally, in the table legends you should avoid abbreviations like CV, HV, etc. - in general, the table should be explained enough that you do not need to go to the text).
